# Genomic and Phenotypic Analysis of *Salmonella enterica* Bacteriophages Identifies Two Novel Phage Species

**DOI:** 10.3390/microorganisms12040695

**Published:** 2024-03-29

**Authors:** Sudhakar Bhandare, Opeyemi U. Lawal, Anna Colavecchio, Brigitte Cadieux, Yella Zahirovich-Jovich, Zeyan Zhong, Elizabeth Tompkins, Margot Amitrano, Irena Kukavica-Ibrulj, Brian Boyle, Siyun Wang, Roger C. Levesque, Pascal Delaquis, Michelle Danyluk, Lawrence Goodridge

**Affiliations:** 1Food Safety and Quality Program, Department of Food Science and Agricultural Chemistry, McGill University, Montreal, QC H9X 3V9, Canada or sudhakar.bhandare@nottingham.ac.uk (S.B.);; 2School of Veterinary Medicine and Science, University of Nottingham, Nottingham LE12 5RD, UK; 3Canadian Research Institute for Food Safety, Department of Food Science, University of Guelph, Guelph, ON N1G 2W1, Canada; lawal@uoguelph.ca; 4Institute for Integrative Systems Biology (IBIS), Laval University, Québec, QC G1V 0A6, Canadarclevesq@ibis.ulaval.ca (R.C.L.); 5Faculty of Land and Food Systems, University of British Columbia, Vancouver, BC V6T 1Z4, Canada; siyun.wang@ubc.ca; 6Agriculture and Agri-Food Canada, Summerland, BC V0H 1Z0, Canada; 7Food Science and Human Nutrition Department, University of Florida, Gainesville, FL 32611, USA

**Keywords:** *Salmonella*, bacteriophage, diversity, host specificity, comparative phylogenomics, food safety

## Abstract

Bacteriophages (phages) are potential alternatives to chemical antimicrobials against pathogens of public health significance. Understanding the diversity and host specificity of phages is important for developing effective phage biocontrol approaches. Here, we assessed the host range, morphology, and genetic diversity of eight *Salmonella enterica* phages isolated from a wastewater treatment plant. The host range analysis revealed that six out of eight phages lysed more than 81% of the 43 *Salmonella enterica* isolates tested. The genomic sequences of all phages were determined. Whole-genome sequencing (WGS) data revealed that phage genome sizes ranged from 41 to 114 kb, with GC contents between 39.9 and 50.0%. Two of the phages SB13 and SB28 represent new species, *Epseptimavirus SB13* and genera *Macdonaldcampvirus,* respectively, as designated by the International Committee for the Taxonomy of Viruses (ICTV) using genome-based taxonomic classification. One phage (SB18) belonged to the *Myoviridae* morphotype while the remaining phages belonged to the *Siphoviridae* morphotype. The gene content analyses showed that none of the phages possessed virulence, toxin, antibiotic resistance, type I–VI toxin–antitoxin modules, or lysogeny genes. Three (SB3, SB15, and SB18) out of the eight phages possessed tailspike proteins. Whole-genome-based phylogeny of the eight phages with their 113 homologs revealed three clusters A, B, and C and seven subclusters (A1, A2, A3, B1, B2, C1, and C2). While cluster C1 phages were predominantly isolated from animal sources, cluster B contained phages from both wastewater and animal sources. The broad host range of these phages highlights their potential use for controlling the presence of *S. enterica* in foods.

## 1. Introduction

Non-typhoidal *Salmonella* (NTS) is the leading cause of foodborne illness in Canada [1]. Globally, NTS causes approximately 80 million foodborne-related illnesses and 155,000 deaths each year [2]. Non-typhoidal Salmonellosis is caused by the main species of the genus, *Salmonella enterica*, which consists of six subspecies (I, II, IIIa, IIIb, IV, and VI), with subspecies I containing more than 1500 serotypes with high genetic diversity [3]. In Canada, there have been several large national and international outbreaks of Salmonellosis linked to vegetables, including whole onions, peaches, frozen corn, and cantaloupes, since 2020 [4,5,6,7]. These outbreaks have been caused by several *S. enterica* serotypes (Enteritidis, Newport, Soahanina, Sundsvall, Oranienburg) and have highlighted the need for improved approaches to control the presence and growth of diverse *S. enterica* in fresh and processed fruits and vegetables, which increasingly contribute to the burden of foodborne disease.

Bacteriophages (phages) are increasingly being recognized as natural antimicrobials that can reduce the growth and survival of foodborne pathogens (including *Salmonella*) during food production [8,9,10,11,12] because of their ability to kill their host bacteria [13,14] and also due to the fact that phages can exhibit broad host ranges [15], making them useful for controlling diverse bacterial species such as *S. enterica* [16]. The majority of phages described in the scientific literature appear to be generally host-specific, infecting a subset of species, strains, or serotypes due to the specificity of their host receptors [17]. While some phages can infect a broad range of bacteria belonging to different serotypes, species, and/or genera [17,18], information on these phages is limited.

While *Salmonella* phages and their genomic sequences have been well documented [17,19,20,21,22,23,24,25], the phenotypic and genotypic diversity of *Salmonella enterica* means that there are likely many additional phages with unique features that remain to be characterized. Understanding the biological and genomic characteristics of these phages is essential to the development of phage-based antimicrobial methods to control this foodborne pathogen [13,26]. In this study, we determined the host range spectra and performed comparative genomic and phylogenetic analyses, as well as morphological characterization, of eight *S. enterica* phages isolated from wastewater obtained from the Jean R. Marcotte wastewater treatment plant in Montreal, QC, Canada.

## 2. Materials and Methods

### 2.1. Bacterial Host Strains and Their Growth

Non-typhoidal *S. enterica* isolates (n = 43) representing 30 serotypes, and commonly associated and/or previously implicated in outbreaks involving fresh produce, were obtained from the *Salmonella* foodborne Syst-OMICS database (SALFOS, Laval University, QC, Canada, https://salfos.ibis.ulaval.ca/, assessed on 22 January 2023) (Figure 1). Frozen stocks of the isolates were maintained at −80 °C and were revived by streaking them on Luria–Bertani (LB) agar plates followed by overnight incubation at 37 °C. For all experiments, a single colony of a respective isolate from a fresh LB agar plate was inoculated into 10 mL of LB broth and incubated overnight at 37 °C with shaking at 150 RPM.

### 2.2. Bacteriophage Isolation and Propagation

Bacteriophage isolation was carried out using pre-treated sewage sludge samples (1 L) collected from the Jean-R. Marcotte Wastewater Treatment Plant (WWTP) in Montreal by adopting standard enrichment and agar overlay techniques [27] with slight modifications, as described elsewhere [28]. Twelve *S. enterica* isolates, originally isolated from food plants/fresh produce and representing 11 serotypes, were used for phage isolation (Figure 1). These serotypes included Bareilly, Braenderup, Enteriditis, Heidelberg, Infantis, Javiana, Montevideo, Newport, Saintpaul, Thompson, and Typhimurium.

A sewage sample (1 L) was transported to the laboratory on ice, where it was clarified by centrifugation at 10,000× *g* for 30 min in 50 mL tubes and filtered through 0.22 µm syringe filter units (Pall Corporation, Port Washington, NY, United States). A total of 30 mL of filtrate was mixed with 9 mL of 5× Rappaport-Vassiliadis (RV), 1 mL of 1 M CaCl_2_, and 10 mL of an overnight culture of a *Salmonella* isolate, and the suspension was incubated in a 250 mL flask at 37 °C for 48 h while shaking at 50 RPM. The culture was clarified by centrifugation at 10,000× *g* for 10 min and the supernatant was filtered through a 0.22 µm filter to remove bacterial debris. An aliquot (10 µL) of filtrate was spot-inoculated on a lawn of the *Salmonella* isolate used for isolation and the plate was incubated at 37 °C overnight. The next day, the lawns were checked for zones of lysis or individual plaques. Plaque purification was carried out multiple times using a streak plating method, as previously described [29], to obtain a clonal phage isolate. Phage stocks were concentrated by centrifugation (40,000× *g* for 2 h at 4 °C) and stored at 4 °C until further use. For routine use, phages were propagated by the liquid lysate method [28] where a mid-exponential-phase bacterial host culture was infected with phage and incubated overnight. The next day, the phage and the bacterial host culture was centrifuged (10,000× *g* for 10 min) and filtered through a 0.45 µm syringe filter, followed by determination of the phage titer.

### 2.3. Host Range Profiles

Host ranges for each of the isolated phages were determined using the agar overlay method. Following the preparation of agar overlays (as described above), the lawns (prepared for each of the 43 *Salmonella* isolates to be tested) were spot-inoculated with aliquots (10 μL) of each phage, with titers of 10^8^ plaque-forming units (PFUs)/mL. The spots were allowed to dry before being incubated overnight at 37 °C. The scoring for lysis was completed as reported elsewhere [30,31], where 0 indicated no lysis and +3 indicated complete clear lysis.

### 2.4. Phage DNA Isolation, Sequencing, and Annotation

Genomic DNA from the *Salmonella* phages was extracted using the Wizard DNA Clean-Up system (A7280; Promega, Madison, WI, USA) following the modified Promega Wizard method, as described by the Center for Phage Technology, Texas A&M University, USA [32]. Extracted DNA was purified by ethanol precipitation [33] and whole-genome sequencing (WGS) was performed on an Illumina MiSeq platform (Illumina, San Diego, CA, USA) with 300 bp paired-end libraries and 30× coverage. Raw sequence reads were assembled using the A5 pipeline [34] and genome annotation was completed using the Bacterial and Viral Bioinformatics Resource Center (formerly PATRIC) [35,36]. Annotations were manually curated and the coding sequences (CDSs) were used to interrogate the NCBI database using BLASTP (https://blast.ncbi.nlm.nih.gov/Blast.cgi?PAGE=Proteins (accessed on 6 January 2024)) [37]. An HHpred search of the Pfams database was used to identify conserved protein motifs [38]. To assign a protein to a gene sequence, at least 90% identity was sought in BLASTP searches for protein motifs [39]. Based on the presence or absence of a gene encoding integrase, phages were putatively classified as temperate or virulent, respectively [26].

### 2.5. Phylogenetic and Comparative Genomic Analyses

The whole-genome alignments of the phages reported in this study and 113 homologs that were extracted from NCBI were generated using MAFFT v7.453 [40]. Maximum likelihood trees were constructed using IQtree v2.2 (https://github.com/Cibiv/IQ-TREE) [41] and visualized using Microreact (https://microreact.org/ (accessed on 6 January 2024)) [42]. Phages were assessed for genes encoding antimicrobial resistance, virulence, and type I-VI toxin–antitoxin modules using CARD [43], VFDB [44,45], and TADB [46], respectively. Likewise, phage genome sequences were screened for the presence of tailspike proteins using an in-house manually curated custom database. Whole-genome comparisons and their visual representations were carried out using EasyFig (https://github.com/mjsull/Easyfig) and iTOL (https://itol.embl.de) [47].

### 2.6. Electron Microscopic Imaging of the Phages

Phages were purified for electron microscopy by equilibrium density gradient centrifugation through CsCl at ≈22,000 RCF for 24 h in a Beckman Ultra centrifuge (TL100) [48]. Post centrifugation, the residual CsCl was removed from the phage fraction by centrifuging 500 μL of the supernatant fluid through an Amicon Ultra-0.5 30 kDa MWCO centrifugal filter unit (Millipore Ltd., Burlington, MA, USA). Following purification, transmission electron microscopy was conducted at the Imaging—Microscopy Platform of the Institute of Integrative Biology and Systems (IBIS), Laval University, Quebec City, QC, Canada.

## 3. Results

### 3.1. Phage Isolation and Biological Characterization by Host Range Profile

Eight phages (Table 1) were isolated using *S. enterica* isolates representing serotypes commonly associated with fresh produce outbreaks. The host range profile of the isolated phages was determined using 43 different *Salmonella* isolates from 30 serotypes (Figure 1 and Figure 2). *Salmonella* isolates representing the top eight plant-associated salmonellosis-causing serotypes were given importance due to their frequent implication in produce-associated outbreaks, which account for the majority of foodborne outbreaks [49,50,51,52]. The top eight serotypes were identified as serotypes Newport, Javiana, Enteriditis, Typhimurium, Thompson, Heidelberg, Saintpaul, and Poona (Figure 2). One additional serotype, Litchfield, was also included because it has been implicated in numerous outbreaks associated with melons [53]. The broadest host range was exhibited by phages SB3 and SB6, followed by SB9. Phages SB3 and SB6 lysed 88.3% (n = 38/43) isolates, while phage SB9 lysed 86% (n = 37/43) isolates, phage SB10 lysed 76.7% (n = 33/43), phages SB13 and SB18 lysed 83.7% (n = 36/43) isolates, phage SB28 lysed 81.3% (n = 35/43), and phage SB15 showed the lowest lysis percentage of 67.4% (n = 29/43) isolates. Five of eight (62.5% (SB3, SB6, SB9, SB10, SB13)) phages isolated in this study lysed the top eight serotypes and Litchfield (Figure 2).

### 3.2. Comparative Phylogenomic Analysis of Phage Understudy

To ascertain the suitability of the isolated phages for biocontrol purposes, the phages were sequenced. Raw reads were assembled into draft genomes that were functionally annotated as described above (see Section 2).

#### 3.2.1. Comparative Gene Content Analysis of Phages under Study

The genome size of the phages ranged from 41 to 114 kb, while their GC contents were between 39.9 and 50.0% (Table 1). Whole-genome-based phylogenetic analysis of the eight phages revealed two clusters (cluster A and B) and two singletons (Figure 3). Phages SB28, SB9, SB10, and SB13 were nested together in cluster A, SB15 and SB3 were clustered in cluster B, and SB6 and SB18 were singletons. The homology of phage genomes with previously identified phages was determined using a comparative genomic approach with BlastN. Phages in cluster A were heterogenous in terms of their genome size, number of tRNAs, and coding sequences (Table 1). Phages SB9 and SB10 were similar to *Salmonella* phage 116 (accession number: NC_048007.1; 99.58% identity and 88% coverage) and *Salmonella* phage fuchur (accession number: NC_048869.1; 97% identity and 96% coverage), which were both isolated from wastewater in Denmark [54] (Figure 4A). Phages SB13 and SB28, respectively, had 99% (coverage: 92%) and 96% (coverage: 79%) nucleotide similarity to *Escherichia* phage Supergirl (accession number: MZ501105.1), which was isolated from a sewage plant in Switzerland (Figure 4B) [55], and *Salmonella* phage E1 (accession number: NC_010495.1), which was isolated from an unknown source in the United Kingdom (Figure 4C) [56].

Phages SB3 and SB15 (cluster B) had a comparable genome size (41 kB), number of tRNAs, and GC content, and shared 99% nucleotide similarity between them. They also had >95% nucleotide identity (coverage: 97%) with *Salmonella* phage vB_SenS_AG11 (accession number: NC_041991.1), which was isolated from sewage in Guelph, Canada, in 2007 [57,58] (Figure 4D). The singletons, Phage SB6 and SB18, respectively, had 99.35 and 98% nucleotide sequence similarity to *Salmonella* phage oselot (accession number: NC_048871.1; 95% coverage), recovered from wastewater in Denmark [59], and *Erwinia* phage phiEa21-4 (accession number: NC_011811.1; 98% coverage), which was isolated in soil beneath a pear tree with active blight in Canada [60,61] (Figure 4E,F). Collectively, given the broad host ranges of the phages isolated in this study, these results confirm the utility of wastewater as a rich source from which to isolate diverse and broad-host-range phages [62,63].

The genomes of the phages isolated in this study were screened for the presence of lysogeny, antimicrobial resistance, virulence, or toxin–antitoxin-related genes. Of note, genes encoding virulence, type I-VI toxin–antitoxin modules, antimicrobial resistance, or lysogeny were not detected in any of the phages. Given the broad host range of the isolated phages, an assessment of the genes encoding bacterial recognition proteins was conducted. Three (SB3, SB15, and SB18) out of the eight phages possessed tailspike proteins (Figure 3). The tailspike proteins from phages SB3 and SB15 were identical (100% nucleotide sequence similarity and coverage) and shared 93% nucleotide sequence similarity with the *Salmonella* phage vB_SenS_AG11 tailspike protein (accession number: AF012431.1). Although annotation of the phage genome predicted that these tailspike proteins belong to the *Salmonella* phage P22-like tailspike protein, comparative genomic analysis revealed that they shared only 66% nucleotide similarities with the *Salmonella* phage P22 tailspike protein (accession number: NP_059644.1). Phages SB10, SB13, and SB18 had a bulb-like structure at the base of the tail, typical of phages with tailspike proteins, but there were no genes annotated as having a tailspike protein in their genome. More so, BLASTing these phages’ genomes (SB10, SB13, and SB18) against a manually curated custom database containing 8077 unduplicated tailspike proteins did not yield any significant hits. However, phage SB18 contained a gene annotated as a baseplate protein that had 97% nucleotide similarity to the baseplate spike protein of *Erwinia* phage phiEa21-4 (accession number: YP_004327040.1).

#### 3.2.2. Assessing the Genetic Relatedness of Phages under Study with Other *Salmonella* Phage Genomes

Phylogenetic analysis of the phages showed high genetic relatedness with other phages. For example, based on genome comparisons with previously identified *Salmonella* phages deposited in public databases, the phages were clustered into three main clusters designated as A, B, and C, which were further grouped into seven subclusters (A1, A2, B1, B2, C1, C2, and C3) (Figure 5). The singleton phages SB6 and SB18 (Figure 3) were placed in different subclusters of cluster C. Phage SB6 clustered with Bacteriophage T5-like in subcluster C1, whereas SB18 was found in subcluster C2 along with *Erwinia* phages phiEa21-4 and *Salmonella* phage ST-3, among others. Phage SB28 with *Salmonella* phage E1 formed a distinct subcluster (A1), while phages SB9, SB10, and SB13 were found in subcluster A2 with other *Salmonella* phages (Figure 5). Conversely, phages SB3 and SB15 were clustered in subcluster B1 together with *Salmonella* phage vB_SenS_AG11. In subcluster A2, phage SB9 was clustered with SB10 and SB13 along with *Salmonella* phages S116 and fuchur and *Escherichia* phage Supergirl, respectively (Figure 5).

Of note, the phages that were isolated from animals/animal products were predominantly found in subcluster C1. Relative to the other clusters, cluster B contained phages isolated from more diverse sources and associated with the presence of tailspike proteins. For the phages with *Salmonella* as the host, serotype Typhi was enriched in cluster A (Figure 5). Overall, these results reiterate the high genetic diversity that has been observed among *S. enterica* phages [25].

### 3.3. Salmonella Phages under Study Are Morphologically and Taxonomically Different

Phage morphological characterization was performed using transmission electron microscopy (TEM). Based on the TEM imaging, all of the phages belonged to the *Siphoviridae* morphotype except SB18, which belonged to the *Myoviridae* morphotype (Table 1). *Siphoviridae* morphotypes have long flexible tails, while *Myoviridae* morphotypes are defined by a rigid contractile tail (ICTV) [64,65]. Images of phages representative of clusters A, B, and C are shown in Figure 6A–C, respectively.

In silico analysis of the isolated phages and their comparison with existing homologs allowed for taxonomical associations. With recent changes to the taxonomic classification of phages, major families such as *Siphoviridae*, *Podoviridae*, and *Myoviridae* have been abolished by the International Committee on Taxonomy of Viruses (ICTV [64,65] (https://ictv.global/vmr/current, accessed on 6 January 2024)). Phylogenomic analysis was employed to classify the phages into classes and genera based on the new ICTV classification schema [66]. All of the phages were classified into the class *Caudoviricetes*. The phages SB6, SB9, SB10, and SB13 were classified into the family *Demerecviridae*, being part of the subfamily *Markadamsvirinae* and the genus *Epseptimavirus*, and species-level classification was given only for SB10 and SB13 viz. *Epseptimavirus fuchur* and *Epseptimavirus SB13*, respectively. Phage SB18 belonged to the same family as phages SB6, SB10, and SB13 but was classified as being part of the subfamily *Ounavirinae* and the genus *Kolesnikvirus*. Phage SB28 represents a novel virus that was recently classified by ICTV as *Macdonaldcampvirus* at the genus level and as *Macdonaldcampvirus SB28* at the species level. Phages SB3 and SB15 were classified as being part of the subfamily *Guernseyvirinae* and *Jerseyvirus AG11* at the genus level (https://viralzone.expasy.org/6319, accessed on 6 January 2024).

## 4. Discussion

*Salmonella enterica* is a major foodborne pathogen of global importance and its genomic diversity has been widely studied [67,68,69,70,71,72]. Subspecies I contains more than 1500 of the total 2600+ serotypes in the species and is of utmost importance with respect to human infections [73]. Many studies have reported on the isolation and characterization of *Salmonella* phages, but these studies have reported on the isolation of phages from only a few of the 1500 serotypes from subspecies I, with a specific focus on the serotypes most commonly implicated in human disease [74]. In this study, we isolated *Salmonella enterica* phages from wastewater and assessed their diversity and host specificity using a combination of microscopic, biological, and genomic approaches. Phage isolation was conducted on a panel of 12 *S. enterica* isolates representing 11 serotypes (Figure 1). The isolated phages were then characterized using a panel of 43 highly diverse *S. enterica* isolates representing 30 serotypes (Figure 1 and Figure 2) that are commonly associated with fresh produce outbreaks [75,76,77]. There are only a few reports of phages isolated from many of the serotypes chosen in this study, allowing for an assessment of *Salmonella* phage diversity from food-plant-associated serotypes.

Wastewater is reported to be a rich source of phages, containing a vast diversity of both temperate and virulent phages that infect a wide array of host bacteria, including *Salmonella* [63,78]. In this study, eight *S. enterica* phages were isolated from the Jean-R. Marcotte WWTP in Montreal, which is the third largest WWTP in North America and provides wastewater treatment for the entire island of Montreal [79]. Montreal (population: 4.34 million) is a large urban and diverse city with inhabitants from more than 50 nationalities [80] and a rich history of gastronomy, meaning that a wide variety of North American and ethnic foods are consumed within the city as a whole. Therefore, the size of the Jean-R. Marcotte WWTP and the diversity of wastewater treated there made it an ideal location for the isolation of a diverse group of *S. enterica* phages.

Our results indicate that the phages isolated in this study had broad host ranges, as the majority of the *Salmonella* isolates used in this study were lysed by the phages. The broadest host range phages (SB3, SB6, and SB9) lysed more than 85% of the isolates used for lytic spectra testing, which suggests the presence of one or more conserved receptors used by these phages to infect *S. enterica* and indicates that these phages could be good candidates for a phage-based control strategy for reducing the microbial contamination of food plant produce. In an earlier study, we demonstrated that phages SB3 and SB6 successfully reduced *S. enterica* populations on lettuce and cantaloupe tissues [12]. Future studies will focus on the elucidation of the bacterial receptor/s used by phages SB3, SB6, and SB9 to better assess their potential for use in controlling foodborne contamination due to common and rare *Salmonella* serotypes. In addition, further studies on the potential of other phages isolated in this study to reduce *S. enterica* on food matrices are required to fully assess their biocontrol efficacy.

Whole-genome-based phylogeny of the phages sequenced in this study revealed that two of them were unique, while the remaining six were closely related to previously sequenced phages. The genome size and GC and gene contents of the phages were heterogenous. Comparative genomic analysis showed that, when compared to 113 phages from public databases, the eight phages from this study were grouped into three different clusters: cluster A (SB28, SB9, SB10, and SB13), cluster B (SB3 and SB15), and cluster C (SB6, SB18) (Figure 5). Phages are known to be heterogenous and have been recognized as one of the major drivers of diversity, evolution, and adaption of their hosts in different environmental matrices, including wastewater [17,19,20,21,22,23,24,25,81,82]. *Salmonella* Typhi phages were enriched in cluster A, whereas phages that were isolated from animals/animal products were enriched in cluster C. This diversity is reflected in the host- and/or source-associated clustering of the phages in the phylogeny [20].

The taxonomic classification of phages is based on electron microscopy and whole-genome sequencing [83]. The great majority of the phages sequenced in this study belonged to the *Siphoviridae* morphotype, while one (SB18) belonged to the *Myoviridae* morphotype. Two phages (SB13 and SB28) differed significantly from the previously sequenced phage genomes and represented novel species. Based on this observation, the bacterial and archaeal viruses subcommittee (BAVS) of ICTV has created two new species viz. *Epseptimavirus SB13* and *Macdonaldcampvirus SB28* [84]. The novel genus *Macdonaldcampvirus* has one more species *Macdonaldcampvirus* ViIIE1 attributed to *Salmonella* phage Vi II-E1 (AM491472.1). Knowledge regarding classification aids in the design of phage cocktails for biocontrol purposes, as phages with different morphotypes use different host receptors [85] and help to overcome phage resistance [86]. Indeed, phage cocktails containing different morphotypes could be more effective in reducing bacterial loads in food products. In a previous study, the inclusion of phages SB3 and SB6 (*Siphoviridae* morphotype) from this study with three other phages belonging to the *Myoviridae* morphotype in a five-phage cocktail was effective in reducing *Salmonella enterica* on lettuce and cantaloupe flesh sections [12].

One of the safety concerns of using phages as biocontrol in food products is their propensity to harbor and/or facilitate horizontal gene transfer of antimicrobial resistance determinants in the environment [87,88,89]. In this study, none of the phages carried genes encoding virulence or antimicrobial resistance. Another concern arises from the pharmacological limitations of using phages as antimicrobials. For example, there is a significant size disparity between phage particles and antibiotic and other antimicrobial compounds, with phages being millions of times larger and composed of multiple proteins. This size discrepancy restricts dosing options and diminishes uptake and transportation rates [90]. To address this limitation, interest is increasingly turning to utilizing phage components as antimicrobials, with a specific focus on using phage lysins that are active against Gram-positive bacteria [91]. These enzymes are not active against Gram-negative bacteria due to the protective nature of the outer-membrane protein. More recently, several groups have demonstrated the antimicrobial effects of phage tailspike proteins against Gram-negative bacteria. Phage tailspike proteins are highly thermostable and protease-resistant [92]. They possess carbohydrate depolymerase activity and recognize and cleave components of the lipopolysaccharide (LPS) to position the phage toward a secondary membrane receptor during infection [93]. Ayariga et al. [92] demonstrated that the ɛ34 phage tailspike protein has an enzymatic property as an LPS hydrolase and synergizes with the Vero Cell culture supernatant in killing *Salmonella* Newington. Miletic and colleagues [94] expressed the receptor-binding domain of the phage P22 Gp9 tailspike protein in plant tissue (*Nicotiana benthamiana*) and demonstrated that upon oral administration of lyophilized leaves expressing Gp9 tailspike protein to newly hatched chickens, *Salmonella* concentrations were reduced on average by approximately 1-log, relative to controls. This result is underwhelming but promising as each dose contained much less of the Gp9-ELP. Other studies led to reduced *Salmonella* motility and colonization [94,95,96]. In this study, three phages possessed tailspike proteins, viz. SB3 (GenBank: QBQ74073.1); SB15 (GenBank: QHI00505.1); and SB18 (GenBank: QHI00609.1). Future work will include the isolation and purification of these tailspike proteins and analyze them as potential antimicrobials to control *S. enterica* in foods.

## 5. Conclusions

In order to study *Salmonella enterica* bacteriophage diversity and host specificity, we isolated and characterized eight bacteriophages using microscopic, biological, and genomic approaches. Biological characterization by host range profile revealed that all eight phages were broad-host-range phages; genomically, none of the phages possessed virulence, toxin, antibiotic resistance, or lysogeny genes, and we could classify them using their physical/morphological characterization. Phages SB3 and SB6 had been previously identified and proved to be good biocontrol candidates, owing to their desirable characteristics. Phages SB3, SB6, and SB9 had the broadest host range and could be promising candidates for phage-based biocontrol, either alone or in a cocktail. This study reported two new phage species recognized by ICTV, i.e., *Epseptimavirus SB13* and *Macdonaldcampvirus SB28*. Most of the phage genomes had a significant number of hypothetical proteins and this lack of understanding or the unknown functions of these proteins could be limitations to the use of these (and other) phages as biocontrol agents. Nonetheless, our attempt to understand the diversity and host specificity of the isolated phages could contribute constructively to our understanding of phage biology and help us to better utilize this understanding in the development of biocontrol strategies in controlling *Salmonella* worldwide, in various environmental settings.

## Figures and Tables

**Figure 1 microorganisms-12-00695-f001:**
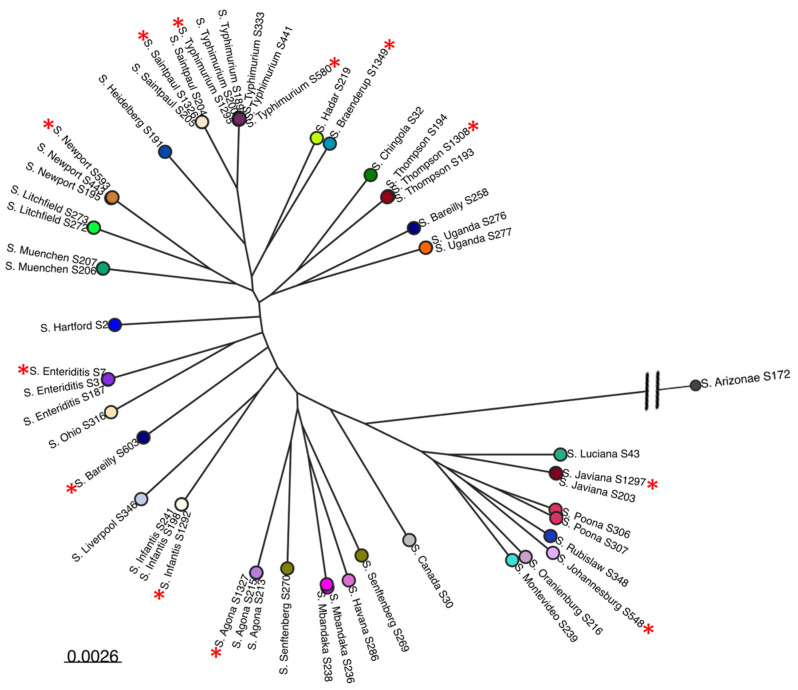
Maximum likelihood tree of 55 *Salmonella* strains representing 30 serovars used for phage propagation and host specificity in this study. The *Salmonella* strains used for propagating the phages in this study are indicated with a red star. Core-genome sequence alignment was generated from draft genomes using MAFFT v7.453 and the tree that was bootstrapped with 1000 replicates for node support was constructed using FastTree v2.1.11. The scale bar at the bottom represents nucleotide substitution per site.

**Figure 2 microorganisms-12-00695-f002:**
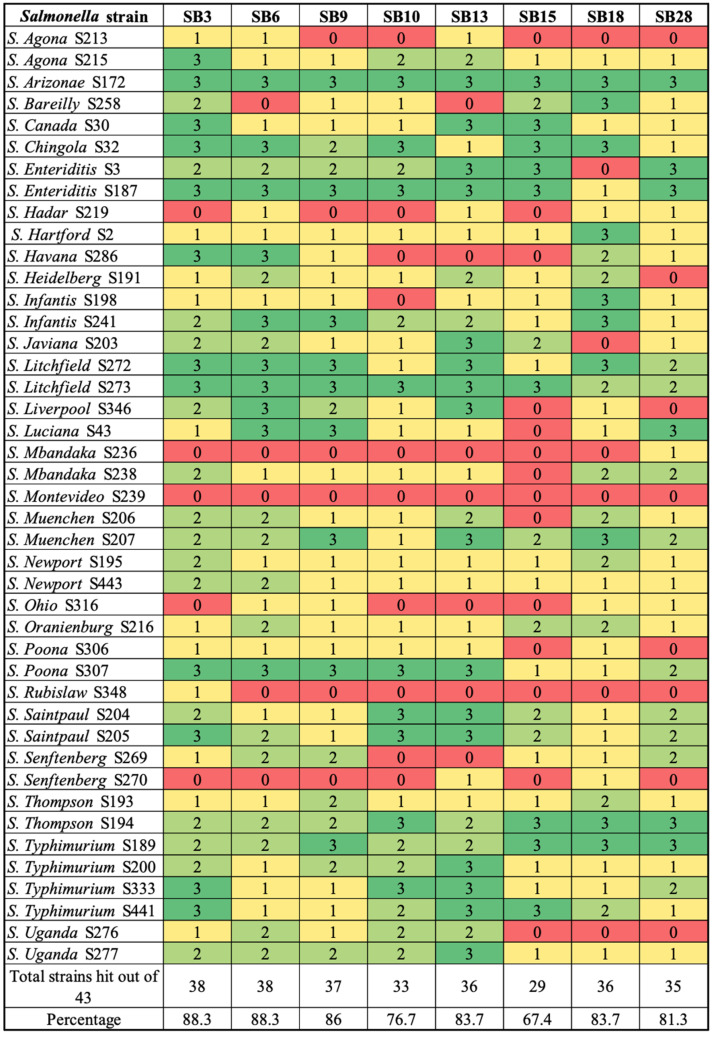
Heat map of the lytic spectra of isolated phages with scoring of lysis, where 0 indicates no lysis and 1 to 3 indicate clearing (3 indicates complete clear lysis).

**Figure 3 microorganisms-12-00695-f003:**
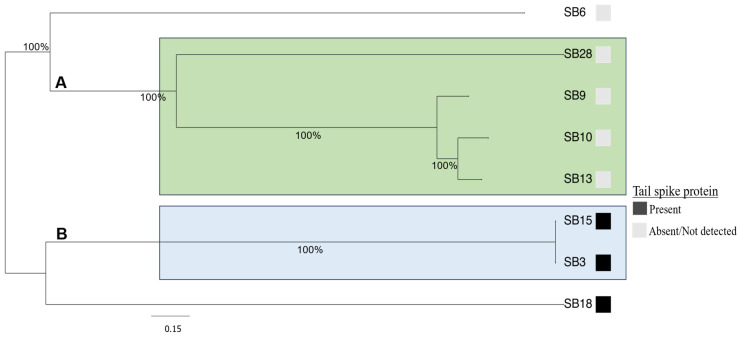
Maximum likelihood phylogenetic tree showing the relatedness of the phages under study. The phage genomes’ alignment was generated using MAFFT v7.453 and the tree that was bootstrapped with 1000 replicates for node support was constructed using IQtree v2.2. The scale bar at the bottom shows nucleotide substitution per site. Phages in the light green rectangle belonged to cluster A; those within the blue rectangle were designated as cluster B phages, and the others were singletons. The gray and white boxes depict the presence or absence of tailspike proteins in the phages.

**Figure 4 microorganisms-12-00695-f004:**
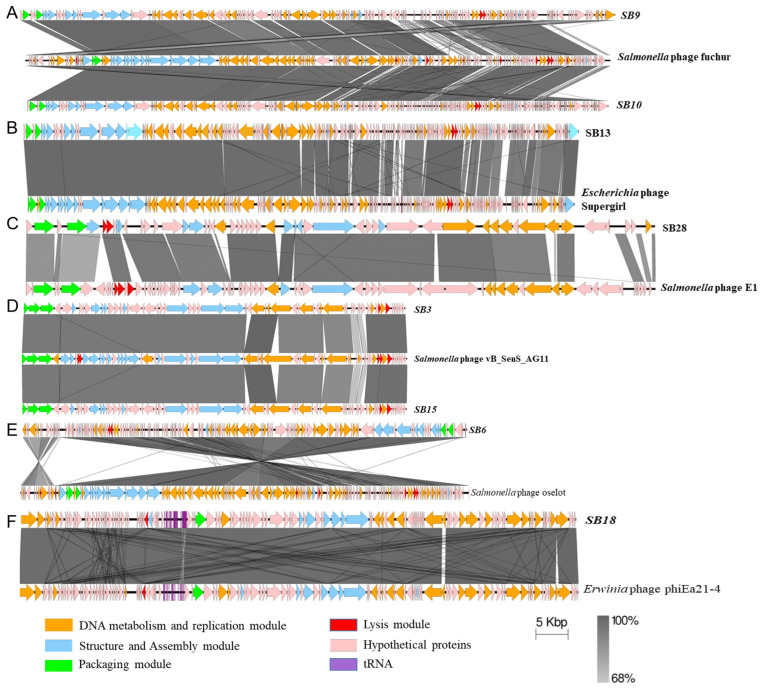
Homology and gene synteny comparison of phages isolated in this study with previously sequenced phages. BlastN comparisons of the phages in this study with their closest references. (**A**) SB9 and SB10 with *Salmonella* phage fuchur; (**B**) SB13 with *Escherichia* phage Supergirl; (**C**) SB28 with the reference *Salmonella* phage E1; (**D**) SB3 and SB15 with *Salmonella* phage vB_SenS_AG11; (**E**) SB6 with *Salmonella* phage oselot, and (**F**) SB18 with *Erwinia* phage phiEa21-4. The direction of arrows indicates the DNA strand direction. Figures were generated using EasyFig.

**Figure 5 microorganisms-12-00695-f005:**
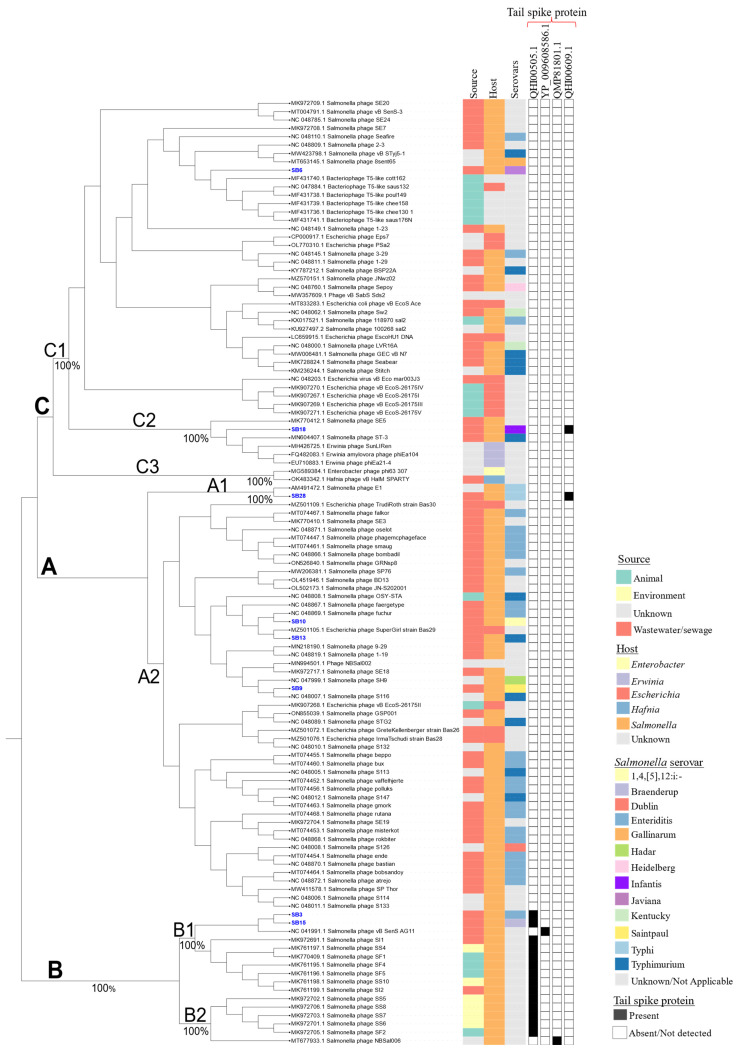
Maximum likelihood phylogenetic tree showing the relatedness of phages under study, with previously identified *Salmonella* phages having a coverage of ≥80% and nucleotide homology of ≥95%. The phage genomes’ alignment was generated using MAFFT v7.453 and the maximum likelihood tree that was bootstrapped with 1000 replicates for node support was constructed using IQtree v2.2. The first three blocks show the source, host, and serovars (for the *Salmonella* host) of the phages, while the other blocks indicate the presence and absence of tailspike proteins. The scale bar at the bottom indicates the nucleotide differences amongst them. The phages under study are labeled in blue.

**Figure 6 microorganisms-12-00695-f006:**
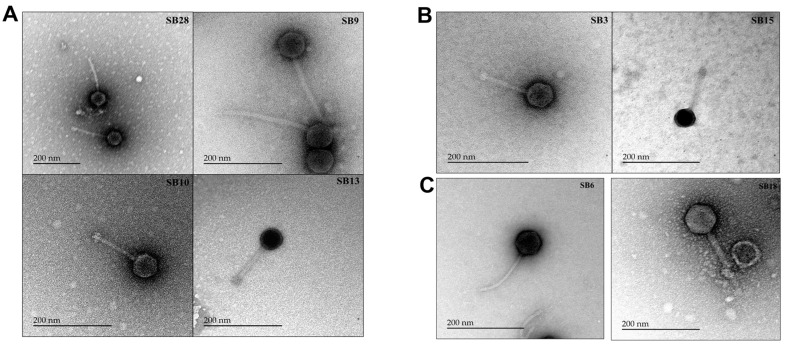
Transmission electron microscopy (TEM) images of representative phages from different clusters. (**A**) Cluster A phages: subcluster A1: SB28; subcluster A2: SB9, SB10, and SB13. (**B**) Cluster B phages: SB3 and SB15. (**C**) Cluster C phages: subcluster C1: SB6; subcluster C2: SB18.

**Table 1 microorganisms-12-00695-t001:** Summary of *Salmonella enterica* bacteriophages isolated in this study.

Phage	*Salmonella* Host Strain Used for Isolation *	Morphotype	New Classification	Old Classification	GenBank Accession Number	Genome Length (Kb)	No. of Coding Genes	No. of tRNA	G+C (%)
SB3	*S. enterica* ser. Enteritidis (S7)	*Siphovirus*	Genus: Jerseyvirus; Species: Jerseyvirus AG11	Genus: Jerseyvirus; Species: *Salmonella* virus AG11	MK578530	41.15	63	0	50.0
SB6	*S. enterica* ser. Javiana (S1297)	*Siphovirus*	Genus: Epseptimavirus	Genus: Haartmanvirus	MK809530	112.31	180	16	39.9
SB9	*S. enterica* ser. Saint-Paul (S1326)	*Siphovirus*	Genus: Epseptimavirus	Genus: Tequintavirus	MK867835	113.98	180	20	39.9
SB10	*S. enterica* ser. Typhimurium (S1295)	*Siphovirus*	Genus: Epseptimavirus; Species: Epseptimavirus fuchur	Genus: Tequintavirus; Species: *Salmonella* virus fuchur	MK947458	111.35	183	21	40.1
SB13	*S. enterica* ser. Typhimurium (S580)	*Siphovirus*	Genus: Epseptimavirus; Species: Epseptimavirus SB13.	Genus: Tequintavirus	MK947459	112.51	175	15	39.9
SB15	*S. enterica* ser. Braenderup (S3)	*Siphovirus*	Genus: Jerseyvirus; Species: Jerseyvirus AG11	Genus: Jerseyvirus; Species: *Salmonella* virus AG11	MK759883	41.44	73	0	50.0
SB18	*S. enterica* ser. Infantis (S43)	*Myovirus*	Genus: Ounavirinae; Species: Kolesnikvirus Ea214	Genus: Ounavirinae; Species: Kolesnikvirus	MK759884	85.31	118	17	43.8
SB28	*S. Typhi* ser. T42 DEF 472 (S203)	*Siphovirus*	Genus: Macdonaldcampvirus; Species: Macdonaldcampvirus SB28	Was not classified beyond *Siphoviridae* family	MK947460	45.13	73	0	46.2

* Numbers in parenthesis indicate Salfos number.

## Data Availability

Data are contained within the article.

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
