# Peer review of "Genomic and Phenotypic Analysis of Salmonella enterica Bacteriophages Identifies Two Novel Phage Species"

_microorganisms, 2024, doi:10.3390/microorganisms12040695_

Round 1
Reviewer 1 Report
Comments and Suggestions for Authors
In the manuscript “Genomic and phenotypic analysis of Salmonella enterica bacteriophages identifies two novel phage species” the authors describe the isolation and characterization (genetic, microscopic, and host range) of eight Salmonella phages. The methods used for the study are correct; the discussion is generally clear. I believe that the manuscript can be of interest for the readers of the journal.
However, I disagree with some of the authors' conclusions and suggest to describe the results more clearly.
1. L.410-412: “Whole genome-based phylogeny of the eight phages in this study revealed the uniqueness and high genetic diversity among them and as well as previously sequenced phages”. Taking into consideration that only two from the eight studied Salmonella phages represent new species (one is a member of a new genus), and the remaining six phages are quite close of known phages, this statement is wrong. Moreover, some of the studied phages are quite close between themselves.
2. L.398-400: “The 398 broadest host range phages (SB3, SB6 and SB9) lysed more than 85 % of the isolates used for lytic spectra testing…”. Compare to L.474: “Phages SB9, SB13, SB18 and SB28 had the broadest host range and could be…”. Please, specify what is correct.
3. L.382-384: “Phage isolation was conducted on a panel of highly diverse S. enterica isolates (Figure 1&2) representing 30 serotypes…”. It is not true. Phage isolation was performed on a panel of 13 isolates. The isolated phages were then characterized using a panel of 30 diverse S. enterica serotypes.
4. L. 96-98: “Thirteen Salmonella isolates, originally isolated from food plants/fresh produce, and representing thirteen serotypes, were used for phage isolation (Figure 1).” Unfortunately, only 12 Salmonella isolates are indicated with red asterisks in Fig.1.
5. The name of the phages in Fig. 3 differs from the names of the phages in other figures. The correct taxonomic names of phages should be given at least once in the text and short names that were used throughout the text should be added in parentheses once.
Figures:
- Figure 4: I understand that it is difficult to depict genomes from 41 kb to 113 kb in one drawing; however, so far the SB28 genome (45 kb) is larger than the S9, SB 10, SB13 genomes (11-113 kb). Please, arrange the figure, align the ABCDF and the beginnings of the genomes to the left.
- Figure 5: The arrows cover the names of the studied phages. Please, correct.
- Figures 6-8: What is the idea of presenting the TEM images on genetic clusters? Just images.
Minor comments.
L57: “Bacteriophages (phages) are …”. It's better to start with a new paragraph.
L59: [8,9–12].
L71: [13,26].
L.359-362: It is an unfinished sentence or just the first part of a sentence that includes lines 362-363.
I would recommend the author to make the corrections.
Comments on the Quality of English LanguageMinor editing of English language required
L.359-362: It is an unfinished sentence or just the first part of a sentence that includes lines 362-363.
In addition, some sentences are difficult to understand.
Reviewer 2 Report
Comments and Suggestions for Authors
Well-written paper. Only a few points requiring clarification. How were the 43 isolates of Salmonella chosen from the 55 listed in Figure 1? What subspecies/serotypes were the 13 isolates used for isolation of the phages? Were any of the Salmonella used outbreak strains? A few more details here would be helpful.
L46 Salmonella is #1 cause of foodborne illness in Canada
Figure 2 - all Salmonella with the same serotype should be listed together. Should arrange serotypes in Alphabetical order as otherwise difficult to find ones of interest.
L188 Some clarification needed here as to criteria distinguishing clusters vs singletons.
L431 Phage cocktail efficacy is not solely determined by morphotype of phage. Always need to evaluate cocktails in vivo - or could have some surprises.
L460 0.75 log reduction is likely not physiologically meaningful - any time < 1 log unlikely to be so. Lysins seem like a good strategy but results so far are bit underwhelming.
